# Variation of the Water Level in the Yangtze River in Response to Natural and Anthropogenic Changes

**Jinxin Liu [1], Jinyun Deng [1,2], Yuanfang Chai [1,2,3], Yunping Yang [2,4,*], Boyuan Zhu [5] and Sixuan Li [1]**

[1] StateKey Laboratory of Water Resources and Hydropower Engineering Science, Wuhan University, Wuhan 430072, China; liujinxin1994@whu.edu.cn (J.L.); 15924072753@163.com (J.D.); yuanfangchai@163.com (Y.C.); sixuandashijie@163.com (S.L.)
[2] Changjiang River Scientific Research Institute, Changjiang Water Resources Commission, Wuhan 430010, China
[3] Department of Earth Sciences, Vrije Universiteit Amsterdam, Boelelaan 1085, 1081 HV Amsterdam, The Netherlands
[4] Key Laboratory of Engineering Sediment, Tianjin Research Institute for Water Transport Engineering, Ministry of Transport, Tianjin 300456, China
[5] Key Laboratory of Water-Sediment Sciences and Water Disaster Prevention of Hunan Province, School of Hydraulic Engineering, Changsha University of Science & Technology, Changsha 410114, China; 15871435411@163.com
[*] Correspondence: yangsan520_521@163.com

**Abstract:** The water level in the Yangtze River has significantly changed due to the effects of varied precipitation and dam operations, which have exerted significant effects on irrigation, navigation, and ecosystems. Based on the measured data and the proposed calculation method, we analyzed the adjustment mechanisms of the seasonal water level in the whole Yangtze River. The results were as follows. During the dry season, the rising precipitation and dam operation both increased the water level in the upper reaches and in the reaches from the Jianli to Datong stations during 1981–2014. Moreover, dam operations were the dominant factor (except at Datong station). In the reaches from the Yichang to Shashi stations, dam operations are the reason for the reduction of the measured water level in the dry season, while the rising precipitation had an opposite influence. During the flood season, dam operations helped to reduce the water level from the upper reaches to the estuary during 1981–2014, while climate variation stresses adversely affected the entire river basin. In the reaches between the Luoshan and Jianli stations, climate variation is the dominant factor for the increased water level during the flood season, while dam operation is the reason for the reduced water level at the other six hydrological stations.

**Keywords:** water level; human activities; climate variation; Yangtze River

## 1. Introduction

Water level fluctuations play a significant role in the persistence and structure of ecosystems [1,2], navigation, and irrigation conditions—even people's security [3,4]. Climate variability, particularly through changes in precipitation, exerts significant effects on water levels by affecting the streamflow in rivers [5]. It has been suggested that the increasing precipitation caused a rise in the total global runoff during 1901–2002, leading to an increase in the water level [6]. Human activities also produced huge variations in the water level. As the largest scale human activities in global rivers, more than 50,000 large dams (higher than 15 m) in the world have been constructed, with a cumulative storage capacity around 7000 to 8300 km³ [7], which has caused riverbed scouring and reduced water levels.

Under the combined effects of climatic and anthropogenic factors, the mechanisms behind varied water levels have become more complex, which is why there have been few efforts to separate the contributions of climatic and anthropogenic factors on changes in water levels.

As the third longest river in the world, the drainage area of the Yangtze River Basin accounts for 18% of China's territorial area, where 33% of the total grain is produced, and 40% of the GDP of China is made [8]. Under the influence global warming, extreme climate events are more and more frequent and have significantly changed the water level [9]. For instance, the river basin suffered from the extreme drought climate in 2006 [10], which reduced the streamflow and sediment load and thus contributed to the lowest water level in history at the Cuntan, Yichang, Songzi, Taiping, and Ouchi hydrological stations [11]. Until now, more than 50,000 dams, including the largest dam in the world (the Three Gorges Dam (TGD)), have been constructed, which has significantly reduced sediment discharge and changed the seasonal runoff distribution [12]. From 2002 to 2015, the cumulative volumes scoured from the basic flow channel and the bankfull channel in the Yichang–Hukou reach were $15.16 \times 10^8$ m$^3$ and $15.88 \times 10^8$ m$^3$, respectively, due to the TGD operation [13], which contributed to a reduction of the low water level. Under the effects of both climatic and anthropogenic factors, changes in the water level of the Yangtze River Basin have become more complex.

Many studies have analyzed variation in trends of the water level along the main stem of the Yangtze River and its possible affecting factors [14,15]. Under the same discharge, the downward trend of the low water level has become a common research topic due to the riverbed erosion in the middle and lower reaches of the Yangtze River [16–18], while the variation trend in the water level during flood season is still unclear [19,20]. During the dry season, normal reservoir operations increase the streamflow downstream from the dams to meet the needs of navigation, irrigation, and combating drought, which can help to increase the water level [21]. In contrast, the sharp decrease in sediment load induced by water and soil conservation projects, sand extraction, and reservoir construction can help to cause riverbed erosion and thus reduce the water level [22]. Under the combined effects of increased streamflow and reduced sediment load, the variation characteristics of the water level become more complex. Similarly, during the flood season, reservoir operations will reduce the streamflow to alleviate the flood pressure downstream from the dams, which can help reduce the water level [23], while riverbed armoring and beach vegetation can contribute to an increase in the water level [24]. The adjustment mechanisms of the water level during the dry and flood seasons in the Yangtze River are complicated; thus, the variation characteristics and causes of the water level need to be further studied. Furthermore, studies on separating the contributions of climatic and anthropogenic factors on changes in the water level are rare. Some studies separated the effects using different hydrologic models but only focused on a single year [25–27]. In reality, climate variation is a long-term, multi-decadal problem, and reservoirs, especially the TGD, function under different flow regulation rules during different years. Thus, it is more reasonable to use a long time series of hydro-meteorological data to estimate the influence of climatic and anthropogenic factors on adjustments in the seasonal water level during the dry and flood seasons.

In order to evaluate the effects of climatic and anthropogenic factors on the changes in water level, a long time series of hydro–meteorological data (1950s–2016) was collected, and the study area was expanded to the whole main stem of the Yangtze River. The main purposes of this process are as follows: (1) To analyze the variation trends in the average water level in the dry and flood seasons during different periods; (2) to analyze the reasons for the variations; and (3) to separate the contributions of climatic and anthropogenic factors. The research findings can improve our understanding of the relationships among human activities, climate variation, and water level changes.

## 2. Study Area, Methods, and Materials

### 2.1. Study Area

The Yangtze River is about 6397 km long with a drainage area of $1.8 \times 10^6$ km$^2$, originating from the Qinghai Tibet Plateau and flowing into the East Sea [28]. The long-term average annual water discharge and suspended sediment load—the fifth and fourth largest worldwide—are 921 km$^3$ and 480 Mt, respectively [29]. Nine hydrological stations are involved in this study (Figure 1a). The whole river basin is separated into eight sub-regions according to the location of each hydrological station (Figure 1b). The areal precipitation of the catchment of each station is calculated based on the weather stations of all sub-regions above the hydrological station. The Yichang and Datong stations are the boundaries of the upper, middle, and lower reaches of the river. The TGD (44 km) is located in the upper reaches above the Yichang station.

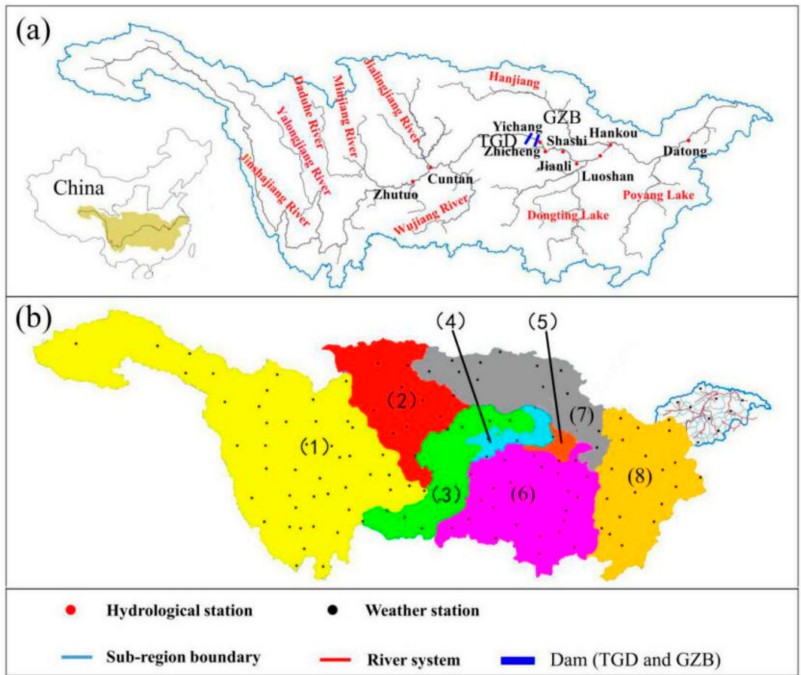

**Figure 1.** (**a**) Geographical position of the Yangtze River Basin, hydrological stations, and the Three Gorges Dam (TGD); (**b**) location of the sub-regions and weather stations.

### 2.2. Data Collection

The daily discharge and water level at the nine hydrological stations during the 1950s–2016 were collected from CWRC (Changjiang River Water Resources Commission). The daily precipitation data for 145 weather stations during 1961 to 2014 (Figure 1b) were collected from the Resource and Environment Data Cloud Platform (http://www.resdc.cn/UserReg.aspx).

### 2.3. Method

#### 2.3.1. Normalization Method

The normalization method [30] can be applied to analyze the variation characteristics of the seasonal runoff distribution. The values of the runoff during the dry season and flood season were changed into dimensionless parameters between 0 and 1 (named "normalized runoff") based on

the ratio of each value of the runoff to the annual runoff (Equations (1) and (2)). The sum of the normalization runoff during the dry season and flood season is 1 (Equation (3)).

$$V_{N.D} = \frac{R_D}{R} \tag{1}$$

$$V_{N.F} = \frac{R_F}{R} \tag{2}$$

$$V_{N.D} + V_{N.F} = 1 \tag{3}$$

where $V_{N.D}$ and $V_{N.F}$ are the normalized runoff during the dry and flood seasons, respectively. $R_D$ and $R_F$ are the runoff during the dry and flood seasons, respectively ($m^3$). R is the annual runoff ($m^3$).

### 2.3.2. Division of the Period

The Mann–Kendall (MK) Test is widely used to check the break points of hydrological data [31,32]. The whole period of the data was thus separated into two parts: the "natural period" (the period with a natural state, before the first break point) and the "impact period" (after the first break point). During the natural period, the impacts of human activities and climate variability can be ignored. Thus, the water level variation is closely related to changes in the streamflow and sediment discharge.

### 2.3.3. Separating the Effects of Climatic and Anthropogenic Factors

Estimating the impacts of climatic and anthropogenic factors on the changes in the water level along the main stem of the river includes three steps:

**Step 1**: Reconstruction of the streamflow in response to climate change (Figure 2a)

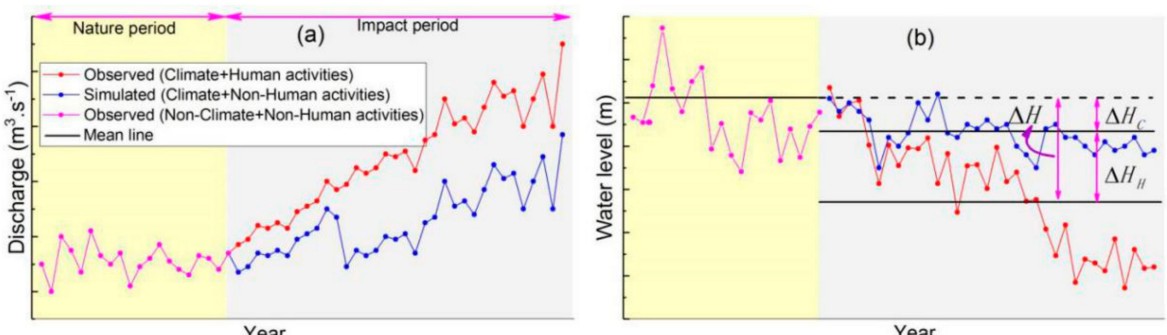

**Figure 2.** A diagrammatic sketch of the calculation method. (**a**) Reconstruction of the streamflow during the dry season (flood season) in response to climate changes (without human activities); (**b**) reconstruction of the water level during the dry season (flood season) in response to climate changes. Note: The red line is the observed discharge (**a**) and water level (**b**); the blue line is the simulated discharge/water level in response to climate change; the pink line is the observed discharge/water level during the period with a natural state.

As one of the most important climate variables, precipitation is widely applied to determine accurate water balance calculations [33]. The relationships between precipitation and runoff during the natural period were built using linear regression analysis. The precipitation during the impact period was put into the regression equations, and thus the streamflow in response to climate variation during the impact period was reconstructed [34,35] (Figure 2a, blue line).

**Step 2**: Reconstruction of the water level in response to climate change (Figure 2b)

Changes in streamflow can have a direct impact on water level. The relationships between the measured runoff and water level during the natural period were built using a linear regression analysis. Then, the reconstructed natural streamflow during the impact period (Step 2) was put into

the regression equations; in this way, the water level in response to climate changes during this period can be simulated (Figure 2b, blue line).

**Step 3**: Estimating the effects of human activities and climate variation

As shown in Figure 2b, the changes in the multi-year average observed water levels during the dry season (flood season) between the natural period and the impact period are defined as $\Delta H$, which can be separated into two parts (Equation (4)): The changes caused by climate variation ($\Delta H_C$) and human activities ($\Delta H_H$). The difference between the simulated water level during the impact period (Figure 2b, blue points) and the measured water level during the natural period (Figure 2b, pink points) illustrates the effects of climate change, while the difference between the simulated and measured water level during the impact period represents the impacts of human activities.

$$\Delta H = \Delta H_C + \Delta H_H \tag{4}$$

## 3. Results

### 3.1. Period Division

As shown in Figure 3, the break points of the normalization runoff during the dry season at the eight hydrological stations are 2001, 2001, 2005, 2004, 2011, 2004, 1994, and 1984. The first break point is 1984, which can be used to separate the whole period. The Gezhouba reservoir, the first dam in the main stem, was commissioned in 1981 [36], which significantly affected the water level downstream from the dam. Finally, the period during 1961–2014 was divided into the natural period (1961–1980) and the impact period (1981–2014) by the year 1980. In order to comprehensively analyze the changes in the water level, the impact period was further divided into three parts: 1981–1990, 1991–2000, and 2001–2014.

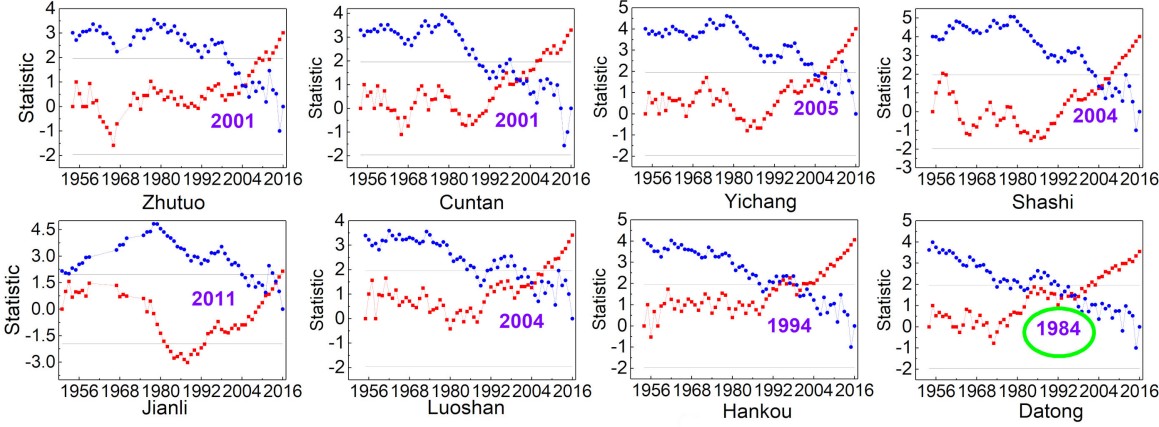

**Figure 3.** Break points of the normalization runoff along the main stem during dry season. Note: The intersection points represent the breaking points.

### 3.2. Variation Trend of Water Level

Figure 4 lists the adjustment mechanisms of the water level along the main stem during the 1950s–2016. We found that the average water level in the dry season at the Yichang and Shashi stations decreased significantly, with a confidence level higher than 99%, while the values at the other six hydrological stations showed an obvious increasing trend. The average water level during the flood season at the Jianli and Luoshan stations showed a rising trend, while the water level at the other six hydrological stations decreased obviously.

Table 1 shows the changes in the multi-year average water level during different periods in comparison with the natural period (1961–1980). At the Yichang and Shashi stations, the decreasing trend of the multi-year average water level during the dry season enhanced with time, with the varied

water level increasing from −0.735 m during 1981–1990 to −1.593 m during 2001–2014, from −0.876 to −2.097 m, respectively. Similarly, the decreased values during the flood season at these two hydrological stations also enlarged over time. At the other six hydrological stations, the multi-year average water level in the dry season during 1981–1990, 1991–2000, and 2001–2014 increased in comparison with the natural period, and the maximum and the minimum increased values were 0.005 m during 1981–1990 and 4.268 m during 2001–2014, both at the Cuntan station. During the flood season, the multi-year average water level at the Jianli, Luoshan, Hankou, and Datong stations increased during 1981–1990 and 1991–2000 and then decreased during 2001–2014.

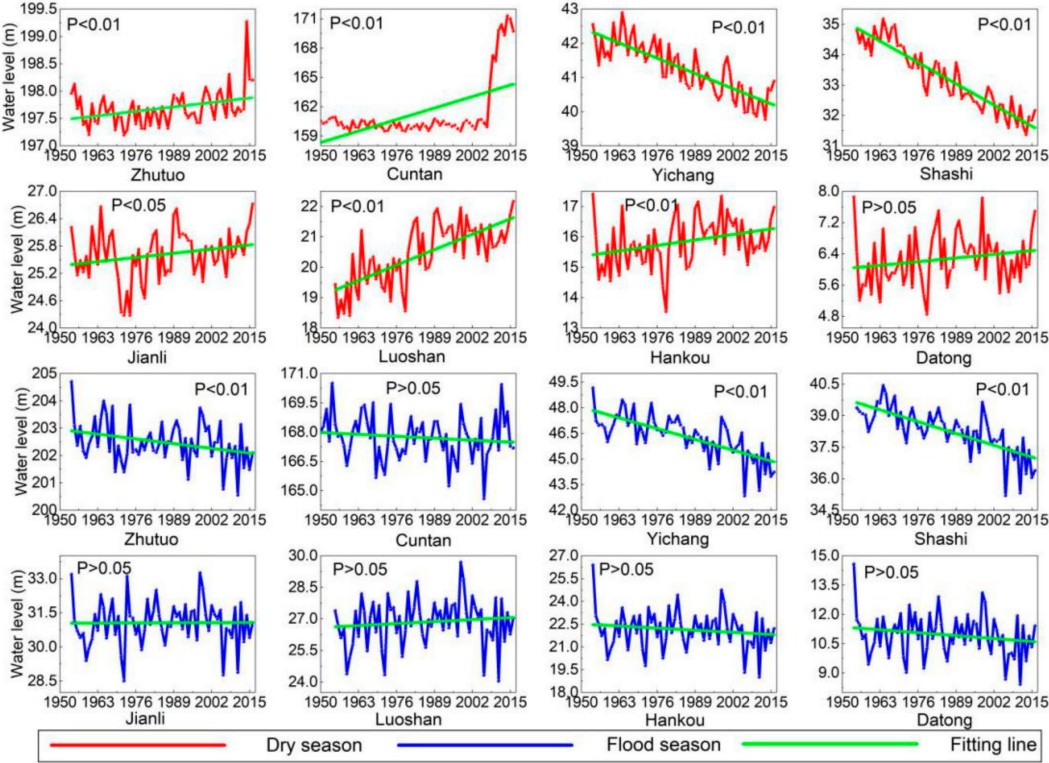

**Figure 4.** Average water level in the dry and flood seasons at the eight hydrological stations during the 1950s–2016. Note: the large increase at the Cuntan station is due to the filling of the TGD.

**Table 1.** Changes in the observed water level in the dry and flood seasons compared with the natural period.

| Station | 1981–2014 | | 1981–1990 | | 1991–2000 | | 2001–2014 | |
|---|---|---|---|---|---|---|---|---|
| | Dry Season | Flood Season | Dry Season | Flood Season | Dry Season | Flood Season | Dry Season | Flood Season |
| Zhutuo | 0.179 | −0.197 | 0.042 | 0.005 | 0.126 | −0.053 | 0.315 | −0.444 |
| Cuntan | 1.782 | −0.014 | 0.005 | 0.281 | 0.077 | −0.334 | 4.268 | 0.004 |
| Yichang | −1.183 | −1.33 | −0.735 | −0.490 | −1.057 | −1.145 | −1.593 | −2.061 |
| Shashi | −1.595 | −1.104 | −0.876 | −0.432 | −1.610 | −0.791 | −2.097 | −1.806 |
| Jianli | 0.333 | 0.140 | 0.355 | 0.212 | 0.318 | 0.515 | 0.328 | −0.179 |
| Luoshan | 1.235 | 0.254 | 1.142 | 0.266 | 1.414 | 0.798 | 1.174 | −0.143 |
| Hankou | 0.691 | −0.033 | 0.714 | 0.148 | 0.905 | 0.481 | 0.521 | −0.529 |
| Datong | 0.389 | −0.107 | 0.603 | 0.058 | 0.550 | 0.460 | 0.122 | −0.629 |

### 3.3. Regression Analysis between the Discharge and Water Level

The regression relationships between the precipitation and discharge in the dry and flood seasons during the natural period are significant, with a confidence level over 95%, except at the Luoshan station (Figure 5), which was used to reconstruct natural runoff during the impact period (Figure 3, blue points).

Figure 6 shows the results of the regression analysis between the water level and the discharge in the dry season and flood seasons during the natural period. It can be seen that the correlations in the whole main stem are significant, with a >99% confidence level, except at the Jianli station. The reconstructed natural runoff during the impact period will be put into the regression equations (Figure 6) to reconstruct the water level in response to climate variation during the impact period.

Table 2 lists the slope values that can represent the influencing degree of the streamflow on the water level. The slope values during the dry season at the eight hydrological stations are all larger than the values during the flood season, which means that the influencing degree of the streamflow during the dry season is more significant than that during the flood season. The cross-section morphologies in the Yangtze River include "U", "V", and "W" [37]. Clearly, with a rising water level, the increase of the cross-sectional area will be increasingly significant, especially during the flood season, which can weaken the effects of the streamflow on the water level during the flood season. The slope values during the dry season and flood season decreased along the main stem of the river in general, from 0.00077 at the Zhutuo station to 0.00024 at the Datong station, from 0.00040 to 0.00018, respectively. Clearly, the influencing degree of the streamflow was reduced along the main stem. The water level behind the TGD reached to 175 m since 2011 [38], while the maximum water level at the Datong station was only 16.64 m on 1 August 1954 [39]. Thus, the impacts of streamflow on the water level reduced along the main stem due to the limited variation range of the water level.

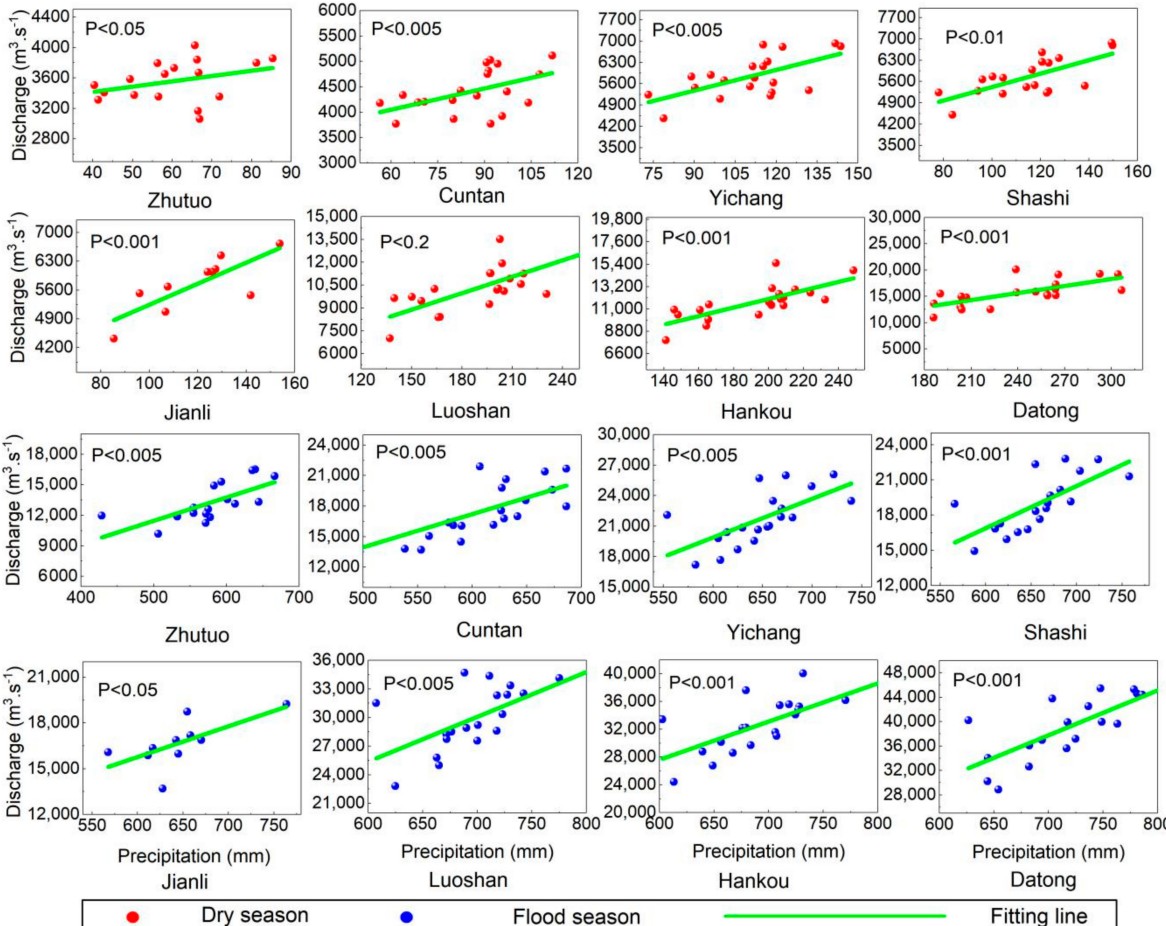

**Figure 5.** Regression analysis between precipitation and discharge in the dry and flood seasons during the natural period (1961–1980).

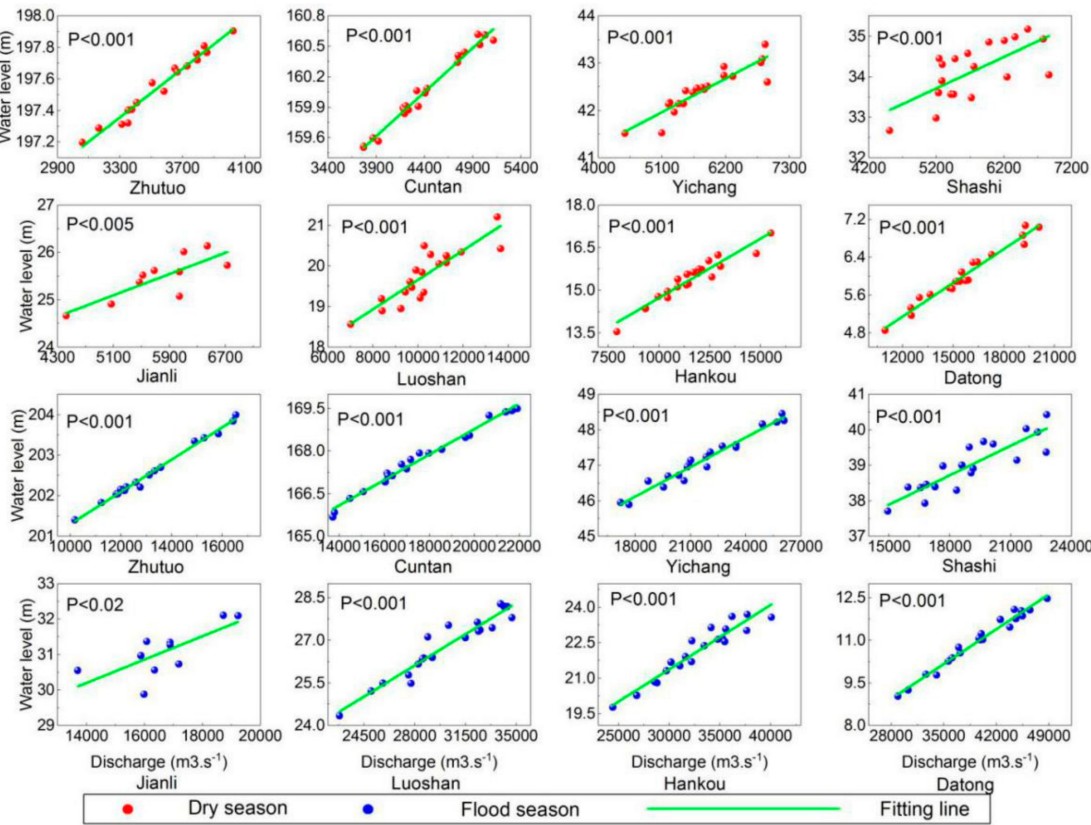

**Figure 6.** Regression analysis between water level and discharge in the dry and flood seasons during the natural period (1961–1980).

**Table 2.** Regression equations and slope values of the water level and discharge. Note: H and Q are the average water level and discharge during the dry season (flood season), respectively.

| Station | Season | Regression Equation | Slope Values |
|---|---|---|---|
| Zhutuo | Dry season | H = 0.00077Q + 194.82 | 0.00077 |
| | Flood season | H = 0.00040Q + 197.32 | 0.00040 |
| Cuntan | Dry season | H = 0.00087Q + 156.22 | 0.00087 |
| | Flood season | H = 0.00045Q + 159.80 | 0.00045 |
| Yichang | Dry season | H = 0.00065Q + 38.17 | 0.00065 |
| | Flood season | H = 0.00028Q + 41.12 | 0.00028 |
| Shashi | Dry season | H = 0.00078Q + 29.68 | 0.00078 |
| | Flood season | H = 0.00028Q + 33.75 | 0.00028 |
| Jianli | Dry season | H = 0.00056Q + 22.24 | 0.00056 |
| | Flood season | H = 0.00033Q + 25.59 | 0.00033 |
| Luoshan | Dry season | H = 0.00036Q + 16.03 | 0.00036 |
| | Flood season | H = 0.00031Q + 17.33 | 0.00031 |
| Hankou | Dry season | H = 0.00042Q + 10.57 | 0.00042 |
| | Flood season | H = 0.00027Q + 13.2 | 0.00027 |
| Datong | Dry season | H = 0.00024Q + 2.29 | 0.00024 |
| | Flood season | H = 0.00018Q + 4.03 | 0.00018 |

*3.4. Effects of Human Activities and Climate Variation during the Dry Season*

3.4.1. Human Activities

Figure 7 shows the observed and the simulated water levels along the main stem, and Table 3 lists the changes in the multi-year average water level during the dry season caused by human activities

($\Delta H_H$) and climate change ($\Delta H_C$). The difference between the observed and the simulated water level during the impact period represents the impacts of human activities. As shown in Figure 7, the simulated water level during the dry season significantly increased at the Yichang and Shashi stations in comparison with the observed water level during 1981–2014 (red lines), while the water level decreased at the other six hydrological stations, especially at the Zhutuo and Cuntan stations. In Table 3, the changes in the water level caused human activities at the Yichang and Shashi stations during 1981–2014 are −1.280 and −1.717 m, respectively, while those at the other six hydrological stations varied from 0.103 m at the Zhutuo station to 1.662 m at the Cuntan station. Thus, human activities reduced the water level at the Yichang and Shashi stations and increased the water level at the other six hydrological stations.

During the periods of 1981–1990, 1991–2000, and 2001–2014, the reduction of the water level caused by human activities at the Yichang and Shashi stations increased from 0.735 m (1981–1990) to 1.710 m (2001–2014) and from 0.876 to 2.247 m, respectively, which means that the effects of human activities on the decreased water level enhanced over time in the reaches between the Yichang and Shashi stations. Moreover, human activities have significantly increased the water level at the Zhutuo and Cuntan stations since 2008 (Figure 7), and the increased water level can reach up to 1.75 m and 11.35 m, respectively, compared to the years of 2007 and 2014.

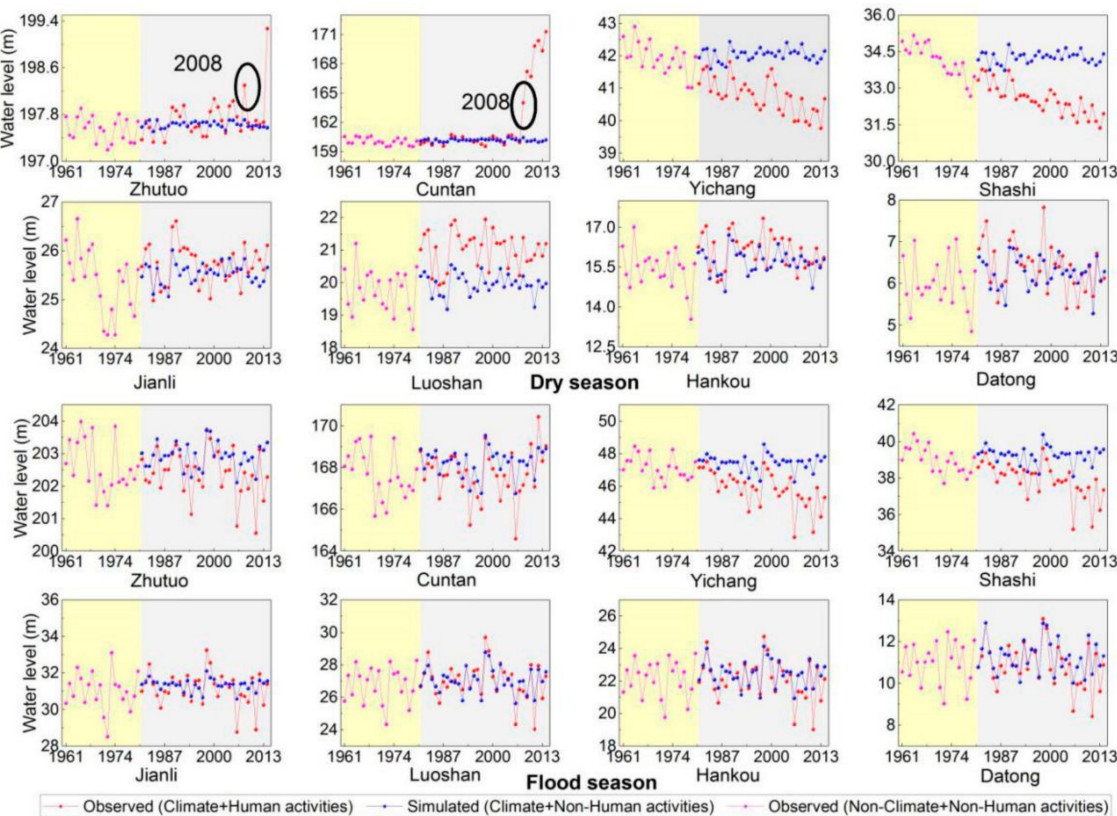

**Figure 7.** Observed and simulated water level in dry and flood seasons during 1961–2014.

**Table 3.** Variation of the multi-year average water level during the dry season caused by human activities ($\Delta H_H$) and climate changes ($\Delta H_C$) and the total change ($\Delta H$).

| Stations | 1981–1990 | | | 1991–2000 | | | 2001–2014 | | | 1981–2014 | | |
|---|---|---|---|---|---|---|---|---|---|---|---|---|
| | $\Delta H$ | $\Delta H_H$ | $\Delta H_C$ | $\Delta H$ | $\Delta H_H$ | $\Delta H_C$ | $\Delta H$ | $\Delta H_H$ | $\Delta H_C$ | $\Delta H$ | $\Delta H_H$ | $\Delta H_C$ |
| Zhutuo | 0.042 | 0.025 | 0.067 | 0.126 | 0.027 | 0.098 | 0.315 | 0.248 | 0.067 | 0.179 | 0.103 | 0.076 |
| Cuntan | 0.005 | 0.079 | 0.083 | 0.077 | 0.072 | 0.149 | 4.268 | 4.144 | 0.124 | 1.782 | 1.662 | 0.119 |
| Yichang | 0.735 | 0.786 | 0.052 | 1.057 | 1.172 | 0.115 | 1.593 | 1.710 | 0.116 | 1.183 | −1.280 | 0.097 |
| Shashi | 0.876 | 0.945 | 0.069 | 1.610 | 1.748 | 0.138 | 2.097 | 2.247 | 0.150 | 1.595 | −1.717 | 0.123 |
| Jianli | 0.355 | 0.240 | 0.115 | 0.318 | 0.148 | 0.147 | 0.328 | 0.147 | 0.182 | 0.333 | 0.174 | 0.159 |

**Table 3.** *Cont.*

| Stations | 1981–1990 | | | 1991–2000 | | | 2001–2014 | | | 1981–2014 | | |
|---|---|---|---|---|---|---|---|---|---|---|---|---|
| | $\Delta H$ | $\Delta H_H$ | $\Delta H_C$ | $\Delta H$ | $\Delta H_H$ | $\Delta H_C$ | $\Delta H$ | $\Delta H_H$ | $\Delta H_C$ | $\Delta H$ | $\Delta H_H$ | $\Delta H_C$ |
| Luoshan | 1.142 | 0.967 | 0.175 | 1.414 | 1.221 | 0.192 | 1.174 | 0.987 | 0.187 | 1.235 | 1.050 | 0.185 |
| Hankou | 0.714 | 0.434 | 0.280 | 0.905 | 0.638 | 0.267 | 0.521 | 0.251 | 0.270 | 0.691 | 0.419 | 0.272 |
| Datong | 0.603 | 0.345 | 0.258 | 0.550 | 0.288 | 0.262 | 0.122 | 0.115 | 0.238 | 0.389 | 0.139 | 0.251 |

### 3.4.2. Climate Variation

The difference between the simulated water level during the impact period and the observed water level during the natural period illustrates the effects of climate variation. As shown in Figure 7, the simulated water level in the dry season during 1981–2014 (blue lines) increased at the eight hydrological stations compared to the observed water level during the natural period (pink lines), which shows that climate variation increased the water level along the main stem. As shown in Table 3, the increased water level caused by climate changes during the whole period (1981–2014) varied from 0.076 m at the Zhutuo station to 0.272 m at the Hankou station. During the periods of 1981–1990, 1991–2000, and 2001–2014, climate variation increased the water level in the dry season during each period, and the effects of these climate changes were enhanced along the main stem as a whole.

### 3.5. Effects of Human Activities and Climate Variation during Flood Season

### 3.5.1. Human Activities

As shown in Figure 7, the simulated water level at the eight hydrological stations increased in comparison with the observed water level during the impact period, which shows that human activities reduced the water level in the whole main stem during 1981–2014. In Table 4, the varied multi-year average water level caused by human activities at the Yichang and Shashi stations ranged up to −1.621 m and −1.364 m, respectively, and that at the other six hydrological stations varied from −0.019 m to −0.569 m. During the periods of 1981–1990, 1991–2000, and 2001–2014, the varied multi-year average water level caused by human activities at the Yichang and Shashi stations increased over time, varying from −0.897 m during 1981–1990 to −2.303 m during 2001–2014 and from −0.815 m to −2.004 m, respectively, which means that the effects of human activities on the decreased water level increased over time. At the other six hydrological stations, human activities reduced the water level during each period, except at the Jianli, Luoshan, and Hankou stations during 1991–2000.

**Table 4.** Variation of the multi-year average water level during the flood season caused by human activities ($\Delta H_H$) and climate changes ($\Delta H_C$) and the total change ($\Delta H$) compared with the natural period (1961–1980).

| Stations | 1981–1990 | | | 1991–2000 | | | 2001–2014 | | | 1981–2014 | | |
|---|---|---|---|---|---|---|---|---|---|---|---|---|
| | $\Delta H$ | $\Delta H_H$ | $\Delta H_C$ | $\Delta H$ | $\Delta H_H$ | $\Delta H_C$ | $\Delta H$ | $\Delta H_H$ | $\Delta H_C$ | $\Delta H$ | $\Delta H_H$ | $\Delta H_C$ |
| Zhutuo | 0.005 | 0.377 | 0.382 | 0.053 | 0.397 | 0.344 | 0.444 | 0.749 | 0.305 | 0.197 | 0.536 | 0.339 |
| Cuntan | 0.281 | 0.432 | 0.713 | 0.334 | 0.688 | 0.355 | 0.004 | 0.582 | 0.586 | 0.014 | 0.569 | 0.555 |
| Yichang | 0.490 | 0.897 | 0.406 | 1.145 | 1.390 | 0.245 | 2.061 | 2.303 | 0.242 | 1.33 | 1.621 | 0.291 |
| Shashi | 0.432 | 0.815 | 0.383 | 0.791 | 1.016 | 0.224 | 1.806 | 2.004 | 0.197 | 1.104 | 1.364 | 0.260 |
| Jianli | 0.212 | 0.208 | 0.420 | 0.515 | 0.200 | 0.315 | 0.179 | 0.472 | 0.293 | 0.140 | 0.197 | 0.337 |
| Luoshan | 0.266 | 0.034 | 0.300 | 0.798 | 0.348 | 0.451 | 0.143 | 0.270 | 0.127 | 0.254 | 0.019 | 0.273 |
| Hankou | 0.148 | 0.278 | 0.427 | 0.481 | 0.056 | 0.425 | 0.529 | 0.723 | 0.194 | 0.033 | 0.363 | 0.330 |
| Datong | 0.058 | 0.275 | 0.332 | 0.460 | 0.055 | 0.515 | 0.629 | 0.704 | 0.075 | 0.107 | 0.387 | 0.280 |

### 3.5.2. Climate Variation

As shown in Figure 7, the simulated water level at each hydrological station during the flood season increased compared to the observed water level during the natural period, which means that climate variation helped increase the water level from the upper reaches to the estuary during 1981–2014. Table 4 shows that the increased water level induced climate changes during 1981–2014

varied from 0.260 m to 0.555 m. During the periods of 1981–1990, 1991–2000, and 2001–2014, climate variation increased the water level in the whole main stem.

## 4. Discussion

### 4.1. Involved Human Activities and Climate Changes

#### 4.1.1. In the Upper Reaches

The TGD carried out 135, 156, and 175 m impoundments (i.e., the water level behind the dam increased to 135, 156, and 175 m, respectively) in 2003, 2005, and 2008, respectively (Zheng, 2016), which did not only affect the water level downstream the dam but also in the upper reaches. During the dry season, the TGD had little influence on the water level at the Zhutuo and Cuntan stations before 2008, while the TGD significantly affected the water level behind the dam after the 175 m impoundment, which is the reason for the sharp rise of the water level during the dry season at these two stations since 2008 (Figure 7). Around 12,994 dams whose corresponding reservoir capacities ranged up to $414.5 \times 10^8$ m$^3$ were constructed in the upper reaches by 2005 [40], which obviously reduced the sediment load and caused riverbed erosion. The first national key project for soil and water conservation was implemented in 1983 [41]; its area was expanded from $156 \times 10^3$ km$^2$ in 1993 to $305 \times 10^3$ km$^2$ in 2012 (Figure 8a), which contributed to riverbed erosion [42]. For the purposes of flood control, hydropower generation, and comprehensive utilization of water resources, the streamflow will be stored behind dams, which can reduce the streamflow in the main stem [28]. Thus, Riverbed erosion and flood peak reduction both reduced the water levels during the flood season at the Zhutuo and Cuntan stations caused by human activities.

#### 4.1.2. In the Middle and Lower Reaches

The operation of more than 50,000 dams has significantly reduced the sediment load in the Yangtze River [43,44], which caused low water channel erosion (Figure 8a), especially in the reaches between the Yichang and Shashi stations [20]. Since 2003, the TGD further reduced the sediment supply to the downstream river channels [45], and the maximum sediment deposition in the reservoir ranged up to $1.96 \times 10^8$ t in 2010 (Figure 8b). In the reaches between the Yichang and Zhicheng stations and between the Zhicheng and Shashi stations, the river bed was significantly scoured (Figure 8c,d), thus decreasing the average elevation of the thalweg in these two reaches up to 4.61 m and 3.65 m during 2002–2015, which helped reduce the water level. Reservoir construction also reduced the streamflow in the main stem, which contributed to the reduced water level. During 1997–1999, the average water storage in the reservoirs was around 57 km$^3$, which significantly increased to 100 km$^3$ during 2000–2012 (Figure 8e). During the flood season, reservoir regulation normally reduced the streamflow to alleviate flood pressure downstream (Figure 8f), and thus the maximum discharge significantly decreased, which helped to reduce the water level (Figure 8g). For instance, according to the regulation scheme of the TGD, the water level behind the dam will increase from 145 m on September 15 to 175 m on 31 October; thus, the stored water volume during this period can range up to $221.5 \times 10^8$ m$^3$ [46]. Furthermore, the population in the river basin increased by over 440 million since 2000, combined with rapid economic development, both of which have created a rising demand for water [47] (Figure 8h) and thus contributed to the reduction of the water level.

In contrast, reservoirs, especially the TGD, will increase the streamflow downstream from the dam to meet the needs of irrigation, shipping, water supply, and the ecological environment during the dry season (Figure 8f) [20]. Moreover, the minimum discharge significantly increased downstream from the dam (Figure 8g). For instance, the TGD operation can regulate river discharge with an increase between 2120 and 1270 m$^3$/s within three to five dry months [48], which can help to increase the water level in this reach [49]. With reduced sediment discharge in the river, riverbed erosion caused an increase in the median size of sediment and bed roughness [50]; both phenomena can help to increase the water level. As shown in Figure 8i,j, the median size of the sediment and bed roughness in the reaches significantly increased. Take the reaches of Yichang-Yidu for instance. The median size of the sediment in 2009 ranged

up to 37.54 mm, which was 59.6 times the value in 2001 (0.63 mm), and the corresponding bed roughness increased from 0.043 to 0.085. Further, the growth of beach vegetation helped to increase the water level during flood season. As shown in Figure 8k, the numbers of days with a discharge larger than 30,000 $m^3 \cdot s^{-1}$ and 40,000 $m^3 \cdot s^{-1}$ (which can both obviously flood beach land) has reduced since the TGD operation, which has helped the growth of beach vegetation and thus increased the water level during the flood season.

In conclusion, during the dry season, the effects of riverbed erosion, increased water usage, and increased water storage in reservoirs were more significant than the impacts of the increased streamflow caused by reservoir operation and increased bed roughness. The consequence of these phenomena is that human activities reduced the water level in the reaches between the Yichang and Shashi stations in the dry season. On the contrary, in the reaches between the Jianli and Datong stations, the effects of the former influencing factors are weaker than the latter's impacts, which caused an increase of the water level in the dry season. During the flood season, the effects of riverbed erosion, increased water usage, increased water storage in reservoirs, and flood peak reduction were more profound than the influence of increased bed roughness and the growth of beach vegetation, which caused a decreased water level in the middle and lower reaches of the river during flood season, induced by human activities.

More important is that human activities increased the water level in the reaches between the Jianli and Hankou stations in the flood season during 1991–2002 (Table 4). The reason for this result might be as follows. As shown in Figure 8l, the reaches were under a condition of deposition during 1991–2002, except in the year of 2002, which helped to increase the water level in the flood season. Furthermore, the flow diversion ratio at the three outlets (Songzi, Taiping and Ouchi) showed a sharp decreasing tread before 2002, which increased the streamflow in the main stem.

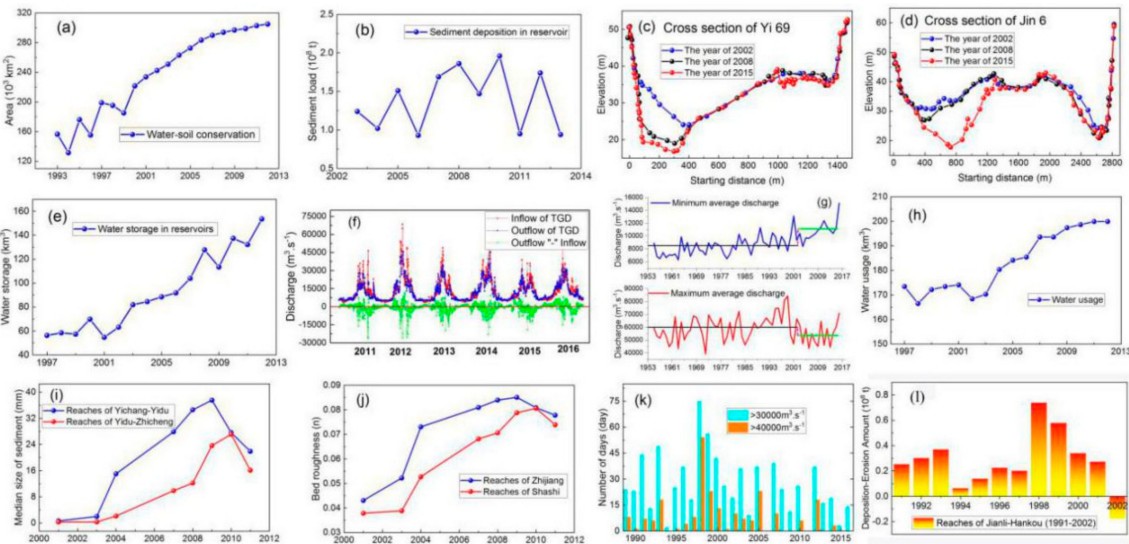

**Figure 8.** Involved human activities in the Yangtze River Basin. (**a**) Area of the water and soil conservation projects in the whole river basin; (**b**) sediment deposition in the TGD; (**c**) evaluation of the cross section of Yi 69 located at the reaches between the Yichang and Zhicheng stations in 2002, 2008, and 2015; (**d**) evaluation of the cross section of Jin 6 located at the reaches between the Zhicheng and Shashi stations in 2002, 2008, and 2015; (**e**) Water storage in reservoirs during 1997–2002; (**f**) inflow and outflow of the TGD; (**g**) minimum and maximum discharges at the Datong station; (**h**) water usage in the whole river basin during 1997–2002; (**i**) variation of the median size of the sediment in the reaches of Yichang-Yidu and Yidu-Zhicheng during 2001–2011; (**j**) variation of bed roughness in the reaches of Yichang-Yidu and Yidu-Zhicheng during 2001–2011; (**k**) number of days with a discharge larger than 3000 $m^3 \cdot s^{-1}$ and 40000 $m^3 \cdot s^{-1}$ at the Yichang station; (**l**) diagrammatic sketch of the effects of beach vegetation on the water level during flood season. Note: The calculation of the bed roughness is based on the equation of n $= \frac{D_{50}^{1/6}}{m}$ [51], where n means the bed roughness, $D_{50}$ means the median size of sediment, and m is the empirical coefficient (selected as 21.5).

### 4.2. Involved Climate Changes

As shown in Figure 9, the precipitation from the upper reaches to the estuary all showed an increasing trend during 1981–2014 in both the dry season and the flood season, which can help to increase the water level in the whole main stem. During the dry season, increasing precipitation and human activities increased the water level both in the upper reaches and in the reaches from the Jianli to Datong stations during 1981–2014, but human activities were the dominant factor (except at the Datong station) (Table 3). In the reaches from the Yichang to Shashi stations, the rising precipitation also helped to increase the water level, while the decreased water level caused by human activities was more significant. Thus, the measured water level at these two stations during 1981–2014 decreased. During the flood season, human activities reduced the water level from the Zhutuo to Datong stations, while the rising precipitation increased the water level in the entire main stem. In the reaches from the Zhutuo to Shashi stations and from the Hankou to Datong stations, the effects of human activities are more obvious than those of climate variation. Thus, the measured water level in the flood season decreased. In contrast, in the reaches between the Luoshan and Jianli stations, climate variation is the dominant factor; thus, the measured water level at these two stations showed an increased trend in the flood season during 1981–2002. Based on the results of the Coupled Model Intercomparison Project phase 5 (CMIP5) during the twenty-first century, the extreme precipitation frequency (maximum 5 day precipitation) will increase by 21% under the RCP8.5 and 11% under the RCP4.5 [52]. This means that the effects of climate change on water levels will be further enhanced in the future especially, during the flood season, which can increase the risks of flood disasters and produce negative impacts on riverbank ecosystems.

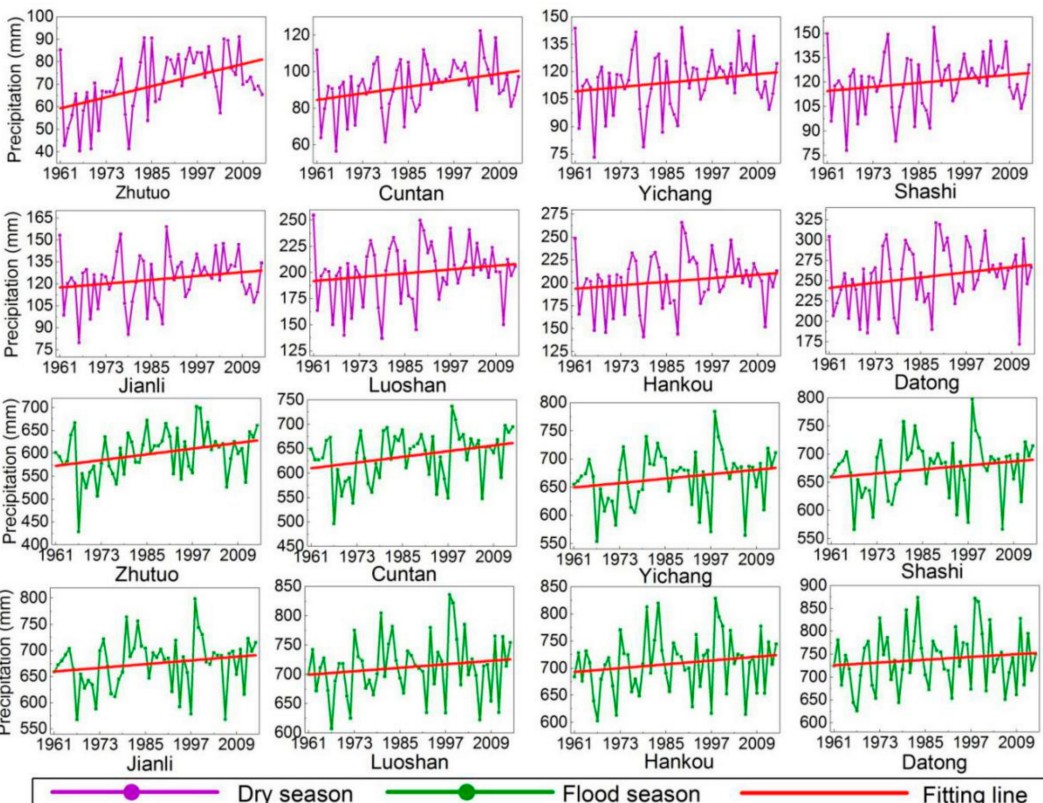

**Figure 9.** Precipitation in the dry season and flood season at eight hydrological stations during 1961–2014.

*4.3. Evaluation of the Errors of Regression Prediction*

Linear regression relationships between precipitation and runoff during the natural period (Figure 5) were built to reconstruct the natural runoff during the impact period. Then, the reconstructed natural runoff was put into the linear regression equations between the runoff and water level during the natural period (Figure 6) to reconstruct the natural water level during the impact period. Although the involved regression relationships were all excellent, with a confidence level over 95% (except at the Luoshan station), the errors of regression prediction still need to be evaluated.

Take the linear regression between the runoff and water level at the Zhutuo station for instance. First, the linear regression relationships for uneven years and even years during the natural period are built; these relationships are used to predict the water level for the same time series. As shown in Table S1, the average absolute errors of uneven years and even years are −0.3 and 0.0 m, and the corresponding average relative errors are only −0.2% and 0.0%. These errors are extremely small, which might be caused by an inherent relation. Thus, to avoid the influence of an inherent relation, the regression equations of uneven years were used to predict the water level of even years, and vice versa. The average ± standard deviation of the absolute errors are −0.3% ± 0.2% and 0.0% ± 0.2%, and the average ± standard deviation of the relative errors are only −0.1% ± 0.1% and 0.0% ± 0.1%, respectively. Thus, the influence of the inherent relation can be neglected. From the upper reaches to the estuary, the maximum average relative errors between the predicted water level and the measured water level are around 1.7%, and the corresponding standard deviation is ± 8.6%. In the whole main stem, the maximum average ± standard deviation of the relative errors between the predicted runoff and the measured runoff is 5.6% ± 11.3%. Thus, the errors, between the predicted water level and the measured water level or between the predicted runoff and measured runoff, are low, which shows that the regression prediction method is reasonable and reliable [53].

*4.4. Effects on Irrigation, Navigation, and Riverbank Ecosystem*

The decreased water levels in the dry season caused by riverbed erosion have precipitated negative effects on irrigation along the main stream, especially in the reaches between the Yichang and Shashi hydrological stations. Since 2003, the TGD operation relieved this pressure by releasing the water behind the dam [27,28]. As a relative value, the waterway depth depends on the water level and riverbed elevation. It has been suggested that increased water depth is higher than the reduced water level (both caused by riverbed erosion), which increased the waterway depth [54]. This decreased water level exposed the beaches of the river channel and lowered the riverbank's elevation, which reduced the habitat ranges of aquatic organisms.

## 5. Conclusions

With intensified human activities and climate variation, the water level has significantly changed from the upper reaches to the estuary. The variation characteristics of the water level in the whole main stem, and its causes, have been analyzed, and the main conclusions are as follows:

(1) During the dry season, human activities, including riverbed erosion caused by reservoir construction and water–soil conservation projects, water storage in reservoirs, and human water consumption, reduced the water level at the Yichang and shashi stations during 1981–2014; the reduced water levels were 1.280 and 1.717 m, respectively. This reduced water level will aggravate the risks of drought disasters and ecosystem degradation. Human activities, including the water supply and increased bed roughness caused by reservoir operation, increased the water level at the other six hydrological stations, and the increased values varied from 0.103 to 1.662 m.

(2) During the flood season, human activities, including riverbed erosion, increased water usage, increased water storage in reservoirs, and flood peak reduction, reduced the water level in the whole main stem during 1981–2014, and the decreased value varied from 0.019 to 1.621 m. This reduced water level during the flood season can alleviate the risks of flood disasters.

(3)     The precipitation from the upper reaches to the estuary showed an increasing trend in both the dry and flood seasons, which can help to increase the water level. During the dry season, the increased water level induced by climate variation varied from 0.076 to 0.272 m and varied from 0.260 to 0.555 m during the flood season. Furthermore, due to the increased extreme precipitation during the 21st century, the effects of climate change on the water level will be enhanced, especially during the flood season, which will need to receive significant attention.

(4)     The 175 m impoundment of the TGD has significantly increased the water level of the dry season at the Zhutuo and Cuntan stations since 2008, and the average increased water level can be up to 0.36 and 3.25 m during 2008–2014.

(5)     In the reaches between Yichang and Shashi stations, human activities significantly reduced the water level in this reach during the dry season, and the effects were enhanced over time.

**Supplementary Materials:** The following are available online at http://www.mdpi.com/2073-4441/11/12/2594/s1.

**Author Contributions:** J.L. and Y.C. conceived the study and wrote the draft of the manuscript. Y.Y., J.D., B.Z. and S.L. contributed to the improvement of the manuscript. Y.C. Prepared Figures 1–9 and Tables 1–4. All authors reviewed the manuscript.

**Funding:** This study was supported CRSRI Open Research Program (Program SN: CKWV2019727/KY; CKWV2018463/KY); National Natural Science Foundation of China [Nos. 41601275; 51809131; 41601275]. We also highly appreciate the valuable insights from the reviewers.

**Acknowledgments:** We highly appreciate the valuable insights from the reviewers and the data provided by the CWRC (Changjiang River Water Resources Commission).

**Conflicts of Interest:** The authors declare that they have no competing interests.

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
