# Peer review of "Variation of the Water Level in the Yangtze River in Response to Natural and Anthropogenic Changes"

_water, doi:10.3390/w11122594_

Round 1

Reviewer 1 Report

Overall, the paper is interesting and brings novelty to the literature. However, some aspects can be improved:

the paper should be professionally checked for Emglish as there are some language flows, typos errors etc abstract should better results of the paper in the study area section, you should present also the future climate change projections for the region also, there is no clear what kind of human interventions significantly affect the water level, so these should be presented since the beginning (including water extraction for agrculture, industry, domestical use etc) instead of nature period, please think about using another term e.g. no significant human intervention or natural state. It is less likely that in the period after 1950s the river is in nature period, meaning no human interventions Fig 2 - should have a legend explaining the two colors The conclusion section is quite weak, you should clealry outline the implication of your study for river management or future decisions

Author Response

Responds to the reviewer’s comments (reviewer 1)

We are so appreciated for your comments and we have revised our paper according to your comments:  

Comment 1:The paper should be professionally checked for English as there are some language flows, typos errors etc. abstract should better results of the paper. in the study area section, you should present also the future climate change projections for the region also, there is no clear what kind of human interventions significantly affect the water level, so these should be presented since the beginning (including water extraction for agrculture, industry, domestical use etc). instead of nature period, please think about using another term e.g. no significant human intervention or natural state. It is less likely that in the period after 1950s the river is in nature period, meaning no human interventions. Fig 2 - should have a legend explaining the two colors; The conclusion section is quite weak, you should clealry outline the implication of your study for river management or future decisions. 

Answer 1:

Thanks for the comments. The whole manuscript has been polished by the “MDPI English Editing Service”;

We have revised the Abstract to provide a more clear findings of this study; Please see the revision in the Page 1 Line 6-9.

About presenting the future climate change in the Yangtze River, here, we presented the changes in future precipitation in the Yangtze River during twenty-first century from the results of the Coupled Model Intercomparison Project phase 5 (CMIP5), and found that the positive effects of climate changes on the water level will further enhanced in the Future. Please see the revision in the Page 15 Line 377, Page 16 Line 378-382.

We have presented what kind of human interventions involving in this study, which can help readers to know the main content at the first. “The water level in the Yangtze River significantly changed due to the effects of varied precipitation and dam operation.......” Please see the revision in the Page 1 Line 6.

It is truth that after 1950s, there already have human activities in the Yangtze River and the effects increased over time. Here, we changed the “nature period” into “the period with natural state” based on your suggestion. Please see the revision in the Page 4 Line 115.

  In the Figure 2, we have add the text for explaining the two colors.“red line is the observed discharge (a)/water level (b); Blue line is the simulated discharge/water level in response to climate change; Pink line is the observed discharge /water level during the period with natural state......” Please see the revision in the Page 5 Line 146-148.

In the conclusion part, we have added some text about the significance of this study for irrigation, shipping and flood prediction.“Due to the increased extreme precipitation during the 21st century, the positive effects of climate changes on the water level will be enhanced especially during flood season, which need receive high attention......”.  Please see the revision in the Page 17 Line 409-417 and 437-440.

Reviewer 2 Report

This paper deals with the variation of water level in the entire Yangtze River in response to natural and anthropogenic changes. The topic is quite interesting, the paper is well written (except some minor mistakes) and I think that it is appropriate for the journal.

I have a concern: in the abstract it is mentioned "The water level in the Yangtze River significantly changed due to the effects of climatic and anthropogenic factors, which brought huge effects on irrigation, navigation and ecosystems" so I was expecting that the authors discussed some of such effects: how the alterations of the water level in the river affected the irrigation, navigation and ecosystems. However, the paper gives a wide description on the variation of water level in 8 measuring stations in the dry and flood seasons connected with the effects of climate and anthropogenic factors, without commenting convincingly how such variation have impacted the irrigation, navigation and the riverbank ecosystems. Could the authors discuss such impacts along the years?

Another small comments are the following:

line 112: Fig 1c should be 1b Fig. 2: what do the red and blue dots represent?

Author Response

Responds to the reviewer’s comments (reviewer 2)

We are so appreciated for your comments and we have revised our paper according to your comments:  

Comment 1: I have a concern: in the abstract it is mentioned "The water level in the Yangtze River significantly changed due to the effects of climatic and anthropogenic factors, which brought huge effects on irrigation, navigation and ecosystems" so I was expecting that the authors discussed some of such effects: how the alterations of the water level in the river affected the irrigation, navigation and ecosystems. However, the paper gives a wide description on the variation of water level in 8 measuring stations in the dry and flood seasons connected with the effects of climate and anthropogenic factors, without commenting convincingly how such variation have impacted the irrigation, navigation and the riverbank ecosystems. Could the authors discuss such impacts along the years?

Answer 1:

Greatly appreciate for these comments. We have add a section to try to provide a better understanding about the effects of the varied water level on irrigation, navigation and ecosystems. “The decreased water level in dry season caused by riverbed erosion have brought negative effects on the irrigation along the mainstream, especially in the reaches between Yichang and Shashi hydrological stations.Since 2003, the TGD operation relieved this pressure by releasing the water behind the dam. As a relative value, waterway depth depends on the water level and the riverbed elevation. It has been suggested that the increased water depth is higher than the reduced water level both caused by riverbed erosion, which increased the waterway depth.The decreased water level exposed the beaches of river channel and lowed the riverbank elevation, which reduced the habitat ranges of aquatic organisms......”. Please see the revision in the Page 17 Line 409-417.

Here, we also analyzed the effects of the future climate change on the water level and the correspondent negative effects based on the results of the Coupled Model Intercomparison Project phase 5 (CMIP5) during twenty-first century. “The extreme precipitation frequency (maximum 5-day precipitation) will increase 21% under the RCP8.5 and 11% under the RCP4.5. This mean that the positive effects of climate changes on the water level will further enhanced in the Future especially in flood season, which can increased the risks of flood disasters and bring negative impacts on riverbank ecosystems......”. Please see the revision in the Page 15 Line 377, Page 16 Line 378-382.

We also enhance the effects of the changed water level on flood disasters and drought disasters in the Conclusion part. Please see the revision in the Page 17 Line 418-440.

Comment 2: line 112: Fig 1c should be 1b Fig. 2: what do the red and black dots represent?

Answer 2: Thanks for this comments. We have revised the “Fig 1c” into “Fig 1b”. Please see the revision in the Page 3 Line 87. Red and black dots respectively represent the “hydrological stations (Please see Figure 1a)” and “weather stations (Please see Figure 1b)”.

Reviewer 3 Report

Review of  water-638981

Variation of water level in the entire Yangtze River in response to

natural and anthropogenic changes

This is an interesting paper on a topic that is relevant in any regulated river, and particularly as a summary of the effects in one of the major water courses with high human impact. It is in the scope of the journal and the study seems to be well carried out. I do have some general comments on the current manuscripts and some suggestion for the authors. These are listed below:

I find the general intro (l34-51) a bit unfocused, and the chosen references seems a bit scattered. I suggest to tightening it up and focus it more towards the general trends that is relevant for your manuscript.

You state that recently, few efforts have been done to study water levels. Does this mean that there were more such studies in the past? Or should “recently” be removed. Is this really true, don´t you find a number of such studies in the ecohydraulics literature?

Generally, I think the intro would benefit from a thorough review both for structure and language, see below.

What is meant with “variation trends”, is it variation in trends or would just trends mean the same?

L102-: The computation of the precipitation for each hydrological station is not clear. Do you talk about the areal precipitation of the catchment of the station?

L127: Isn´t Mann Kendall a trend test and not a break-point analysis? You could of course use this to investigate trend changes and thereby identify break points, but it seems here that you predefine the break points by defining the “nature period” and “impact period”. I do not find any results from Mann Kendall analysis in the results section, how is this used?

Explain how you define the break points in figure 2. I think this part of the manuscript also mixes some results into the method section.

The regression part is a bit confusing, what is seen on figure 3? Is this a result or something else? I did find the actual results nicely presented in the result section, but I think this could have been clarified.

When you on L174 states that you see a significant decreasing trend, what is the significance level? I can see no test results in the text or on the figure.

Figure 4: Is the fitting line based on regression? What is the trend here according to Mann Kendall? What happened at Cuntan between 2002 and 2015? I assume the changes in table 4 are averages or are they changes in trends?

Table 1 should be kept on one page, but I guess this will be handled in the typesetting.

Are the numbers on L180-81 trend line factors?

Table 2: Explain what the coefficient is

When you use the regression method to estimate climatic impact, how sure are you that the relationship will hold in a changing climate (outside the range it was originally made for)?

L271-273: Clarify. Are the values here averages? When you say “varying from 0.897 to 2.303” does this mean that this is the range of variation in these two periods or is it the mean that changed from 0.897 in the first period to 2.303 in the last period? I think it is seen in table 4, but please be specific in the text too. In table 4 these numbers are negative, why are they positive in the text? How did you check the significance over time? Did you test for significance?

L364: What is the comprehensive coefficient?

Conclusion (4): Why the comparison to single years? Can more general statements be concluded?

I do think the paper needs a proper language review.

L53: …where 33% of the total grain is produced and 40% of the GDP of Chinais made?

L68-70: This sentence is heavy to read. “variation trends of the high water level still exists debates” – rephrase, meaning is not clear.

L71: “..combating drought”

L72: I assume it should be “.conservation projects” and not “..conservtion projections”

L74: “Under the both effects..” is not correct.

L81: rarely is probably better than seldom, it should be studies about separating….are rarely

L85&89: Why big “D” in data

L90: Whole mainstream Yangtze river.

L100: Fig. 1a not Figs. 1a

L118: I guess it should be “normalized runoff”? You use that term later.

L132: “..the eight hydrological..”

L195: It sounds like it is the Luoshan station (the only non-significant) one that is used to reconstruct the natural runoff. I suppose this is not what you meant.

L: Several places I think it should be “the natural period” and “the impact period”

L290: insert “did” between which and not

L284: Should mainstream be mainstem, isn´t it more precise to use “mainstem Yangtze”? I think you should consider this as a general change.

L304: I think you could just write dams rather than dams´operation. Check out this term – is it standard to use the “´” after dams?

L338: I think it should be “more significant than the increased streamflow…. and increased bed…”. “influencing” is not necessary and impacts from would also be better.

L409: Intensified

And so on.

Author Response

Responds to the reviewer’s comments (reviewer 3)

We are so appreciated for your comments and we have revised our paper according to your comments:  

Comment 1: I find the general intro (l34-51) a bit unfocused, and the chosen references seems a bit scattered. I suggest to tightening it up and focus it more towards the general trends that is relevant for your manuscript.

Answer 1: Thanks this suggestion. We changed this part, and removed the text that had no strong relationships with this paper. Please see the revision in the Page1 Line 31-34.

Comment 2: You state that recently, few efforts have been done to study water levels. Does this mean that there were more such studies in the past? Or should “recently” be removed. Is this really true, don´t you find a number of such studies in the ecohydraulics literature?

Answer 2: Thanks for this comment. It is not appropriate to use “recently” here. We have removed this word. There do exist many manuscripts about the effects of human activities on the changes in water level, especially hydraulic works. But, about separating the contributions of the effects of human activities and climate changes on the varied water level, the exist few literature. We did not express the ideas that we wand to show clearly. Thus, we have revised the manuscript. “......Few efforts have been made to separate the contributions of climatic and anthropogenic factors on the changes in water level......”. Please see the revision in the Page1 Line 31-34.

Comment 3: What is meant with “variation trends”, is it variation in trends or would just trends mean the same?

Answer 3: Thanks for this comments. We mean variation in trends. We have revised the manuscript. Please see the revision in the Page 2Line 49.

Comment 4:L102-: The computation of the precipitation for each hydrological station is not clear. Do you talk about the areal precipitation of the catchment of the station?

Answer 4: Yes. The computation of the precipitation is about the areal precipitation of the catchment of the station. We have revised to provide a better understanding. Please see the revision in the Page 3 Line 87.

Comment 5: L127: Isn´t Mann Kendall a trend test and not a break-point analysis?  Explain how you define the break points in figure 2.

Answer 5: Truly, the Mann Kendall method was normally used to investigate the trends in time series. The statistic values (as shown in the below figure) can also used to check the break points. (The intersection points in the figure represent the breaking points). The detailed information you can find in the below reference (in the section 3.1)

Yuan, Y. J., Zeng, G. M., Liang, J., Huang, L., Hua, S. S., Li, F., Zhu, Y., Wu, H.P., Liu, J.Y., He, X.X., He, Y. Variation of water level in Dongting Lake over a 50-year period: implications for the impacts of anthropogenic and climatic factors. J. Hydrol2015, 525, 450-456. DOI: 10.1016/j.jhydrol.2015.04.010

Comment 6:You could of course use this to investigate trend changes and thereby identify break points, but it seems here that you predefine the break points by defining the “nature period” and “impact period”. I do not find any results from Mann Kendall analysis in the results section, how is this used? I think this part of the manuscript also mixes some results into the method section.

Answer 6: Greatly appreciate for this comments. We mixed some results into the method section. We have revised the manuscript, and also add the mixed contend into the results part Please see the revision in the Page 5Line 150-158, and “As shown in Fig. 3, the break points of the normalization runoff during dry season at eight hydrological stations.......” Please see the revision in the Page 5Line 160-161

Comment 7: The regression part is a bit confusing, what is seen on figure 3? Is this a result or something else? I did find the actual results nicely presented in the result section, but I think this could have been clarified.

Answer 7: Thanks for this comment. The Figure 3 (Now is Figure 2) is a conceptual figure without involving the results. We show this Figure in order to help readers to get a better understanding of the method. So, in the X-axis and Y-axis, we did not added any data.

Comment 8: When you on L174 states that you see a significant decreasing trend, what is the significance level? I can see no test results in the text or on the figure.

Answer 8: Thanks for this comment. We have added the confidence level in the Figure. Please see the revision in the Page 6 Line 180-181.

Comment 9: Figure 4: Is the fitting line based on regression? What is the trend here according to Mann Kendall? What happened at Cuntan between 2002 and 2015? I assume the changes in table 4 are averages or are they changes in trends?

Answer 9: It is the fitting line based regression. We investigated the confidence level when we built the regression relationships. The trend is based on the regression results; Since 2003, the largest hydraulic work (the Three Gorges Dam) was implemented, which stored plenty of water and significantly increased the water level in the upper reaches of the dam, including the Cuntan hydrological station; The values in the Table 4 are the averages;

Comment 10: Table 1 should be kept on one page, but I guess this will be handled in the typesetting.

Answer 10: Thanks for this suggestion. We have kept the Table on one page. .

Comment 11: Are the numbers on L180-81 trend line factors?

Answer 11: The numbers are the varied water level by comparing different periods. To reduce the confusion, we have revised the manuscript. “...... time, with the varied water level increased from -0.735 m during 1981-1990 to -1.593 m during 2001-2014.....”. Please see the revision in the Page 5 Line 171-172.

Comment 12: Table 2: Explain what the coefficient is

Answer 12: We changed the “coefficients” into “Slope values” to improve the understanding. Please see the revision in the Page 7 Line 195, 202 and Table 2. 

Comment 13: When you use the regression method to estimate climatic impact, how sure are you that the relationship will hold in a changing climate (outside the range it was originally made for)?

Answer 13: Before using the regression method to estimate climatic effects, we first evaluate the confidence level of the regression relationships between precipitation and water discharges, and the regression relationships between water level and water discharges. We found that the confidence level was great, especially for the regression relationships between water level and water discharges (all higher than 99% except at Jianli hydrological stations). The high confidence level of the regression relationships can improve the prediction accuracy and hold the relationships in a changing climate. Besides, we also estimate errors of this regression method, and found that the average + standard deviation of the absolute errors are merely -0.3+0.2% and 0.0+0.2%. The errors are small.

Comment 14:L271-273: Clarify. Are the values here averages? When you say “varying from 0.897 to 2.303” does this mean that this is the range of variation in these two periods or is it the mean that changed from 0.897 in the first period to 2.303 in the last period? I think it is seen in table 4, but please be specific in the text too. In table 4 these numbers are negative, why are they positive in the text? How did you check the significance over time? Did you test for significance?

Answer 14: The values in this section are the multi-year average values during different period. To reduce the confusion, we further make it clear about the meanings of the values in the text. Please see the revision in the Page 12 Line 261-262. 

In the table, the values are negative, but this text we used the word “reduction”, so we used the positive values in the text. To reduce the confusion, we changed the negative values in the text all into positive values. Please see the revision in the Page 12 Line 266. 

Actually, we did not check the significance over time. So, we changed the words “significance over time” into “increased over time”. From the Table 4, we can see that the effects of the human activities on the decreased water level truly enhanced over time. Taking the Zhutuo station for instance, the contributions of the human activities on the water level are -0.377, -0.397 and -0.749 during 1981-1990, 1991-2000 and 2001-2014 respectively. Please see the revision in the Page 12 Line 268.

Comment 15:L364: What is the comprehensive coefficient?

Answer 15: It should be the “empirical coefficient”. We have revised the manuscript. Please see the revision in the Page 15 Line 362. 

Comment 16:Conclusion (4): Why the comparison to single years? Can more general statements be concluded?

Answer 16: Thanks for this comment. We have deleted the comparison of single years, and added a more general comparison during the period since the the 175 m impoundment of the TGD. We have revised the manuscript: “The 175 m impoundment of the TGD significantly increased the water level of dry season at Zhutuo and Cuntan stations since 2008, and the average increased water level can be up to 0.36 m and 3.25 m during 2008-2014 respectively......”. Please see the revision in the Page 18 Line 442-444. 

Comment 17:I do think the paper needs a proper language review.

L53: …where 33% of the total grain is produced and 40% of the GDP of China is made?

L68-70: This sentence is heavy to read. “variation trends of the high water level still exists debates” – rephrase, meaning is not clear.

L71: “..combating drought”

L72: I assume it should be “.conservation projects” and not “..conservtion projections”

L74: “Under the both effects..” is not correct.

L81: rarely is probably better than seldom, it should be studies about separating….are rarely

L85&89: Why big “D” in data

L90: Whole mainstream Yangtze river.

L100: Fig. 1a not Figs. 1a

L118: I guess it should be “normalized runoff”? You use that term later.

L132: “..the eight hydrological..”

L195: It sounds like it is the Luoshan station (the only non-significant) one that is used to reconstruct the natural runoff. I suppose this is not what you meant.

L: Several places I think it should be “the natural period” and “the impact period”

L290: insert “did” between which and not

L284: Should mainstream be mainstem, isn´t it more precise to use “mainstem Yangtze”? I think you should consider this as a general change.

L304: I think you could just write dams rather than dams´operation. Check out this term – is it standard to use the “´” after dams?

L338: I think it should be “more significant than the increased streamflow…. and increased bed…”. “influencing” is not necessary and impacts from would also be better.

L409: Intensified

Answer 17: Greatly appreciate for all the above comments. We have revised the manuscript according to all the above detailed comments. Besides, the whole manuscript has been polished by the English service of the “MDPI”.

Round 2

Reviewer 3 Report

I wish to thank the authors for the thorough response to my original comments. I do think the study now is fine and I would recommend it for publication. I have a couple of minor questions:

L374: "This means that the positive effects...." I am not sure if positive is the right word here, particularly since the positive effects on the water level leads to increased flood disasters. I suggest rewriting it to say something like "This means that the effects of climate change on water levels will be further enhanced ...." 

Maybe you could add to the caption of figure 4 that the large increase at the Cuntan station is due to the filling of Three Gorges. The trend line in this plot is a bit out of place since it seems that you have no trend and then a sudden jump. So this is not a trend in the normal use of the word, but rather a sudden change in the underlying process generating the data.

Author Response

Comment 1: L374: "This means that the positive effects...." I am not sure if positive is the right word here, particularly since the positive effects on the water level leads to increased flood disasters. I suggest rewriting it to say something like "This means that the effects of climate change on water levels will be further enhanced ...." 

 Answer 1: Thanks. We have revised this sentence and also checked this problem in the whole manuscript. Please see the revision in Page 17 Line 394, and Page 19 Line 452.

Comment2: Maybe you could add to the caption of figure 4 that the large increase at the Cuntan station is due to the filling of Three Gorges. The trend line in this plot is a bit out of place since it seems that you have no trend and then a sudden jump. So this is not a trend in the normal use of the word, but rather a sudden change in the underlying process generating the data

 Answer 1: Thanks. We have added the caption of figure 4. Please see the revision in Page 7 Line 190.
